# Feature-distributed sparse regression: a screen-and-clean approach

**Jiyan Yang**[†]    **Michael W. Mahoney**[‡]    **Michael A. Saunders**[†]    **Yuekai Sun**[§]

† Stanford University    ‡ University of California at Berkeley    § University of Michigan

jiyan@stanford.edu    mmahoney@stat.berkeley.edu

saunders@stanford.edu    yuekai@umich.edu

## Abstract

Most existing approaches to distributed sparse regression assume the data is partitioned by samples. However, for high-dimensional data ($D \gg N$), it is more natural to partition the data by features. We propose an algorithm to distributed sparse regression when the data is partitioned by features rather than samples. Our approach allows the user to tailor our general method to various distributed computing platforms by trading-off the total amount of data (in bits) sent over the communication network and the number of rounds of communication. We show that an implementation of our approach is capable of solving $\ell_1$-regularized $\ell_2$ regression problems with millions of features in minutes.

## 1 Introduction

Explosive growth in the size of modern datasets has fueled the recent interest in distributed statistical learning. For examples, we refer to [2, 20, 9] and the references therein. The main computational bottleneck in distributed statistical learning is usually the movement of data between compute notes, so the overarching goal of algorithm design is the minimization of such communication costs.

Most work on distributed statistical learning assume the data is partitioned by samples. However, for high-dimensional datasets, it is more natural to partition the data by features. Unfortunately, methods that are suited to such feature-distributed problems are scarce. A possible explanation for the paucity of methods is feature-distributed problems are harder than their sample-distributed counterparts. If the data is distributed by samples, each machine has a complete view of the problem (albeit a partial view of the dataset). Given only its local data, each machine can fit the full model. On the other hand, if the data is distributed by features, each machine no longer has a complete view of the problem. It can only fit a (generally mis-specified) submodel. Thus communication among the machines is necessary to solve feature-distributed problems. In this paper, our goal is to develop algorithms that minimize the amount of data (in bits) sent over the network across all rounds for feature-distributed sparse linear regression.

The sparse linear model is

$$\mathbf{y} = \boldsymbol{X}\beta^* + \boldsymbol{\epsilon}, \tag{1}$$

where $\boldsymbol{X} \in \mathbb{R}^{N \times D}$ are features, $\mathbf{y} \in \mathbb{R}^N$ are responses, $\beta^* \in \mathbb{R}^D$ are (unknown) regression coefficients, and $\boldsymbol{\epsilon} \in \mathbb{R}^N$ are unobserved errors. The model is sparse because $\beta^*$ is $s$-sparse; i.e., the cardinality of $S := \mathsf{supp}(\beta^*)$ is at most $s$. Although it is an idealized model, the sparse linear model has proven itself useful in a wide variety of applications.

A popular way to fit a sparse linear model is the lasso [15, 3]:

$$\widehat{\boldsymbol{\beta}} \leftarrow \arg\min_{\|\beta\|_1 \le 1} \tfrac{1}{2N} \|\mathbf{y} - \boldsymbol{X}\beta\|_2^2,$$

where we assumed the problem is scaled so that $\|\beta^*\|_1 = 1$. There is a well-developed theory of the lasso that ensures the lasso estimator $\widehat{\boldsymbol{\beta}}$ is nearly as close to $\beta^*$ as an oracle estimator $\boldsymbol{X}_S^\dagger \mathbf{y}$, where $S \subset [D]$ is the support of $\beta^*$ [11]. Formally, under some conditions on the Gram matrix $\frac{1}{N}\boldsymbol{X}^T\boldsymbol{X}$, the (in-sample) prediction error of the lasso is roughly $\frac{s \log D}{N}$. Since the prediction error of the oracle estimator is (roughly) $\frac{s}{N}$, the lasso estimator is almost as good as the oracle estimator. We refer to [8] for the details.

We propose an approach to feature distributed sparse regression that attains the convergence rate of the lasso estimator. Our approach, which we call SCREENANDCLEAN, consists of two stages: a screening stage where we reduce the dimensionality of the problem by discarding irrelevant features; and a cleaning stage where we fit a sparse linear model to a sketched problem. The key features of the proposed approach are:

- We reduce the best-known communication cost (in bits) of feature-distributed sparse regression from $O(mN^2)$ to $O(Nms)$ bits, where $N$ is the sample size, $m$ is the number of machines, and $s$ is the sparsity. To our knowledge, the proposed approach is the only one that exploits sparsity to reduce communication cost.
- As a corollary, we show that constrained Newton-type methods converge linearly (up to a statistical tolerance) on high-dimensional problems that are not strongly convex. Also, the convergence rate is only weakly dependent on the condition number of the problem.
- Another benefit of our approach is it allows users to trade-off the amount of data (in bits) sent over the network and the number of rounds of communication. At one extreme, it is possible to reduce the amount of bits sent over the network to $\widetilde{O}(mNs)$ (at the cost of $\log\left(\frac{N}{s \log D}\right)$ rounds of communication). At the other extreme, it is possible to reduce the total number of iterations to a constant at the cost of sending $\widetilde{O}(mN^2)$ bits over the network.

**Related work.** DECO [17] is a recently proposed method that addresses the same problem we address. At a high level, DECO is based on the observation that if the features on separate machines are uncorrelated, the sparse regression problem decouples across machines. To ensures the features on separate machines are uncorrelated, DECO first decorrelates the features by a decorrelation step. The method is communication efficient in that it only requires a single round of communication, where $O(mN^2)$ bits of data are sent over the network. We refer to [17] for the details of DECO.

As we shall see, in the cleaning stage of our approach, we utilize the sub-Gaussian sketches. In fact, other sketches, e.g., sketches based on Hadamard transform [16] and sparse sketches [4] may also be used. An overview of various sketching techniques can be found in [19].

The cleaning stage of our approach is operationally very similar to the iterative Hessian sketch (IHS) by Pilanci and Wainwright for constrained least squares problems [12]. Similar Newton-type methods that relied on sub-sampling rather than sketching were also studied by [14]. However, they are chiefly concerned with the convergence of the iterates to the (stochastic) minimizer of the least squares problem, while we are chiefly concerned with the convergence of the iterates to the unknown regression coefficients $\beta^*$. Further, their assumptions on the sketching matrix are stated in terms of the transformed tangent cone at the minimizer of the least squares problem, while our assumptions are stated in terms of the tangent cone at $\beta^*$.

Finally, we wish to point out that our results are similar in spirit to those on the fast convergence of first order methods [1, 10] on high-dimensional problems in the presence of restricted strong convexity. However, those results are also chiefly concerned with the convergence of the iterates to the (stochastic) minimizer of the least squares problem. Further, those results concern first-order, rather than second-order methods.

## 2 A screen-and-clean approach

Our approach SCREENANDCLEAN consists of two stages:

1. **Screening Stage:** reduce the dimension of the problem from $D$ to $d = O(N)$ by discarding irrelevant features.
2. **Cleaning Stage:** fit a sparse linear model to the $O(N)$ selected features.

We note that it is possible to avoid communication in the screening stage by using a method based on the *marginal* correlations between the features and the response. Further, by exploiting sparsity, it is

possible to reduce the amount of communication to $O(mNs)$ bits (ignoring polylogarithmic factors). To the authors' knowledge, all existing one-shot approaches to feature-distributed sparse regression that involve only a single round of communication require sending $O(mN^2)$ bits over the network.

In the first stage of SCREENANDCLEAN, the $k$-th machine selects a subset $\widehat{S}_k$ of potentially relevant features, where $|\widehat{S}_k| = d_k \lesssim N$. To avoid discarding any relevant features, we use a screening method that has the *sure screening property*:

$$\mathbf{P}\big(\mathsf{supp}(\beta_k^*) \subset \cup_{k \in [m]} \widehat{S}_k\big) \to 1, \tag{2}$$

where $\beta_k^*$ is the $k$-th block of $\beta^*$. We remark that we do not require the selection procedure to be variable selection consistent. That is, we do not require the selection procedure to only selected relevant features. In fact, we permit the possibility that most of the selected features are irrelevant.

There are many existing methods that, under some conditions on the strength of the signal, has the sure screening property. A prominent example is *sure independence screening* (SIS) [6]:

$$\widehat{S}_{\mathsf{SIS}} \leftarrow \{i \in [D] : \tfrac{1}{N}\left|\mathbf{x}_i^T \mathbf{y}\right| \text{ is among the } \lfloor \tau N \rfloor \text{ largest entries of } \tfrac{1}{N}\mathbf{X}^T\mathbf{y}\}. \tag{3}$$

SIS requires no communication among the machines, making it particularly amenable to distributed implementation. Other methods include HOLP [18].

In the second stage of SCREENANDCLEAN, which is presented as Algorithm 1, we solve the reduced sparse regression problem in an iterative manner. At a high level, our approach is a constrained quasi-Newton method. At the beginning of the second stage, each machine sketches the features that are stored locally:

$$\widetilde{\boldsymbol{X}}_k \leftarrow \tfrac{1}{\sqrt{nT}}\mathbf{S}\boldsymbol{X}_{k,\widehat{S}_k},$$

where $\mathbf{S} \subset \mathbb{R}^{nT \times N}$ is a sketching matrix and $\boldsymbol{X}_{k,\widehat{S}_k} \in \mathbb{R}^{n \times d_k}$ comprises the features stored on the $k$-th machine that were selected by the screening stage. For notational convenience later, we divide $\widetilde{\boldsymbol{X}}_k$ row-wise into $T$ blocks:

$$\widetilde{\boldsymbol{X}}_k = \begin{bmatrix} \widetilde{\boldsymbol{X}}_{k,1} \\ \vdots \\ \widetilde{\boldsymbol{X}}_{k,T} \end{bmatrix},$$

where each block is a $n \times d_k$ block. We emphasize that the sketching matrix is identical on all the machines. To ensure the sketching matrix is identical, it is necessary to synchronize the random number generators on the machines.

We restrict our attention to *sub-Gaussian sketches*; i.e., the rows of $\mathbf{S}_k$ are *i.i.d.* sub-Gaussian random vectors. Formally, a random vector $\mathbf{x} \in \mathbb{R}^d$ is 1-sub-Gaussian if

$$\mathbb{P}(\theta^T\mathbf{x} \geq \epsilon) \leq e^{-\frac{\epsilon^2}{2}} \text{ for any } \theta \in \mathbb{S}^{d-1}, \epsilon > 0.$$

Two examples of sub-Gaussian sketches are the standard Gaussian sketch: $\mathbf{S}_{i,j} \overset{i.i.d.}{\sim} \mathcal{N}(0,1)$, and the Rademacher sketch: $\mathbf{S}_{i,j}$ are *i.i.d.* Rademacher random variables.

After each machine sketches the features that are stored locally, it sends the sketched features $\widetilde{\boldsymbol{X}}_k$ and the correlation of the screened features with the response $\widehat{\boldsymbol{\gamma}}_k := \tfrac{1}{N}\boldsymbol{X}_{k,\widehat{S}_k}^T\mathbf{y}$ to a central machine, which solves a sequence of $T$ regularized quadratic programs (QP) to estimate $\beta^*$:

$$\widetilde{\boldsymbol{\beta}}_t \leftarrow \arg\min_{\beta \in \mathbb{B}_1^d} \tfrac{1}{2}\beta^T\widetilde{\boldsymbol{\Gamma}}_t\beta - (\widehat{\boldsymbol{\gamma}} - \widehat{\boldsymbol{\Gamma}}\widetilde{\boldsymbol{\beta}}_{t-1} + \widetilde{\boldsymbol{\Gamma}}_t\widetilde{\boldsymbol{\beta}}_{t-1})^T\beta,$$

where $\widehat{\boldsymbol{\gamma}} = \begin{bmatrix} \widehat{\boldsymbol{\gamma}}_1^T & \ldots & \widehat{\boldsymbol{\gamma}}_m \end{bmatrix}^T$ are the correlations of the screened features with the response, $\widehat{\boldsymbol{\Gamma}} = \tfrac{1}{N}\boldsymbol{X}_{\widehat{S}}^T\boldsymbol{X}_{\widehat{S}}$ is the Gram matrix of the features selected by the screening stage, and

$$\widetilde{\boldsymbol{\Gamma}}_t := \begin{bmatrix} \widetilde{\boldsymbol{X}}_{1,t} & \ldots & \widetilde{\boldsymbol{X}}_{m,t} \end{bmatrix}^T \begin{bmatrix} \widetilde{\boldsymbol{X}}_{1,t} & \ldots & \widetilde{\boldsymbol{X}}_{m,t} \end{bmatrix}.$$

As we shall see, despite the absence of strong convexity, the sequence $\{\widetilde{\boldsymbol{\beta}}_t\}_{t=1}^\infty$ converges q-linearly to $\beta^*$ up to the statistical precision.

---

**Algorithm 1** Cleaning Stage

---

**Sketching**

1: Each machine computes sketches $\frac{1}{\sqrt{nT}}\mathbf{S}_t \boldsymbol{X}_{k,\widehat{S}_k}$ and sufficient statistics $\frac{1}{N}\boldsymbol{X}_{k,\widehat{S}_k}\mathbf{y}$, $t \in [T]$

2: A central machine collects the sketches and sufficient statistics and forms:

$$
\widetilde{\boldsymbol{\Gamma}}_t \leftarrow \frac{1}{nT}\begin{bmatrix} \vdots \\ (\mathbf{S}_t \boldsymbol{X}_{k,\widehat{S}_k})^T \\ \vdots \end{bmatrix} \begin{bmatrix} \cdots & \mathbf{S}_t \boldsymbol{X}_{k,\widehat{S}_k} & \cdots \end{bmatrix}, \quad \widehat{\boldsymbol{\gamma}} \leftarrow \begin{bmatrix} \vdots \\ \frac{1}{N}\boldsymbol{X}_{k,\widehat{S}_k}^T \mathbf{y} \\ \vdots \end{bmatrix}.
$$

**Optimization**

3: **for** $t \in [T]$ **do**

4:      The cluster computes $\widehat{\boldsymbol{\Gamma}}\widetilde{\boldsymbol{\beta}}_{t-1}$ in a distributed fashion:

$$
\widehat{\mathbf{y}}_{t-1} \leftarrow \sum_{k\in[m]} \boldsymbol{X}_{k,\widehat{S}_k}\widetilde{\boldsymbol{\beta}}_{t-1,k}, \quad \widehat{\boldsymbol{\Gamma}}\widetilde{\boldsymbol{\beta}}_{t-1} \leftarrow \begin{bmatrix} \vdots \\ \frac{1}{N}\boldsymbol{X}_{k,\widehat{S}_k}^T \widehat{\mathbf{y}}_{t-1} \\ \vdots \end{bmatrix}.
$$

5:      $\widetilde{\boldsymbol{\beta}}_t \leftarrow \arg\min_{\beta\in\mathbb{B}_1^d} \frac{1}{2}\beta^T \widetilde{\boldsymbol{\Gamma}}_t\beta - (\widehat{\boldsymbol{\gamma}} - \widehat{\boldsymbol{\Gamma}}\widetilde{\boldsymbol{\beta}}_{t-1} + \widetilde{\boldsymbol{\Gamma}}_t\widetilde{\boldsymbol{\beta}}_{t-1})^T\beta$

6: **end for**

7: The central machine pads $\widetilde{\boldsymbol{\beta}}_T$ with zeros to obtain an estimator of $\beta^*$

---

The cleaning stage involves $2T + 1$ rounds of communication: step 2 involve a single round of communication, and step 4 involves two rounds of communication. We remark that $T$ is a small integer in practice. Consequently, the number of rounds of communication is a small integer.

In terms of the amount of data (in bits) sent over the network, the communication cost of the cleaning stage grows as $O(dnmT)$, where $d$ is the number of features selected by the screening stage and $n$ is the sketch size. The communication cost of step 2 is $O(dmnT + d)$, while that of step 4 is $O(d + N)$. Thus the dominant term is $O(dnmT)$ incurred by machines sending sketches to the central machine.

## 3 Theoretical properties of the screen-and-clean approach

In this section, we will establish our main theoretical result regarding our SCREENANDCLEAN approach, given as Theorem 3.5. Recall that a key element of our approach is to prove the first stage of SCREENANDCLEAN establishes the sure screening property, i.e., (2). To this end, we begin by stating a result by Fan and Lv that establishes sufficient conditions for SIS, i.e., (3) to possess the sure screening property.

**Theorem 3.1** (Fan and Lv (2008)). *Let $\Sigma$ be the covariance of the predictors and $\mathbf{Z} = \boldsymbol{X}\Sigma^{-1/2}$ be the whitened predictors. We assume $\mathbf{Z}$ satisfies the concentration property: there are $c$, $c_1 > 1$ and $C_1 > 0$ such that*

$$
\mathbb{P}\big(\lambda_{\max}\big(\tilde{d}^{-1}\widetilde{\mathbf{Z}}\widetilde{\mathbf{Z}}^T\big) > c_1 \text{ and } \lambda_{\min}\big(\tilde{d}^{-1}\widetilde{\mathbf{Z}}\widetilde{\mathbf{Z}}^T\big) < c_1^{-1}\big) \leq e^{-C_1 n}
$$

*for any $N \times \tilde{d}$ submatrix $\widetilde{\mathbf{Z}}$ of $\mathbf{Z}$. Further,*

1. *the rows of $\mathbf{Z}$ are spherically symmetric, and $\boldsymbol{\epsilon}_i \overset{i.i.d.}{\sim} \mathcal{N}(0,\sigma^2)$ for some $\sigma > 0$;*

2. *$\mathsf{var}(\mathbf{y}) \lesssim 1$ and $\min_{j\in S}\big|\beta_j^*\big| \geq \frac{c_2}{N^\kappa}$ and $\min_{j\in S}|\mathsf{cov}(\mathbf{y},\mathbf{x}_j)| \geq \frac{c_3}{\beta_j}$ for some $\kappa > 0$ and $c_2, c_3 > 0$;*

3. *there is $c_4 > 0$ such that $\lambda_{\max}(\Sigma) \leq c_4$.*

*As long as $\kappa < \frac{1}{2}$, there is some $\theta < 1 - 2\kappa$ such that if $\tau = cN^{-\theta}$ for some $c > 0$, we have*

$$
\mathbb{P}(S \subset \widehat{S}_{\mathsf{SIS}}) = 1 - C_2 \exp\big(-\tfrac{CN^{1-2\kappa}}{\log N}\big)
$$

*for some $C$, $C_2 > 0$, where $\widehat{S}_{\mathsf{SIS}}$ is given by (3).*

The assumptions of Theorem 3.1 are discussed at length in [6], Section 5. We remark that the most stringent assumption is the third assumption, which is an assumption on the signal-to-noise ratio (SNR). It rules out the possibility a relevant variable is (marginally) uncorrelated with the response.

We continue our analysis by studying the convergence rate of our approach. We begin by describing three structural conditions we impose on the problem. In the rest of the section, let

$$K(S) := \{\beta \in \mathbb{R}^d : \|\beta_{S^c}\|_1 \leq \|\beta_S\|_1\}.$$

**Condition 3.2** (RE condition). *There is $\alpha_2 > 0$ s.t. $\|\beta\|_{\widehat{\Gamma}}^2 \geq \alpha_1 \|\beta\|_2^2$ for any $\beta \in K(S)$.*

**Condition 3.3.** *There is $\alpha_2 > 0$ s.t. $\|\beta\|_{\widehat{\Gamma}_t}^2 \geq \alpha_2 \|\beta\|_{\widehat{\Gamma}}^2$ for any $\beta \in K(S)$.*

**Condition 3.4.** *There is $\alpha_3 > 0$ s.t. $|\beta_1^T(\widehat{\Gamma}_t - \widehat{\Gamma})\beta_2| \leq \alpha_3 \|\beta_1\|_{\widehat{\Gamma}} \|\beta_2\|_{\widehat{\Gamma}}$ for any $\beta \in K(S)$.*

The preceding conditions deserve elaboration. The cone $K(S)$ is an object that appears in the study of the statistical properties of constrained M-estimators: it is the set the error of the constrained lasso $\widehat{\beta} - \beta^*$ belongs to. Its image under $X_{\widehat{S}}$ is the transformed tangent cone which contains the prediction error $X_{\widehat{S}}(\widehat{\beta}_T - \widehat{\beta}^*)$. Condition 3.2 is a common assumption in the literature on high-dimensional statistics. It is a specialization of the notion of restricted strong convextiy that plays a crucial part in the study of constrained M-estimators. Conditions 3.3 and 3.4 are conditions on the sketch. At a high level, Conditions 3.3 and 3.4 state that the action of the sketched Gram matrix $\widehat{\Gamma}_t$ on $K(S)$ is similar to that of $\widehat{\Gamma}$ on $K(S)$. As we shall see, they are satisfied with high probability by sub-Gaussian sketches. The following theorem is our main result regarding the SCREENANDCLEAN method.

**Theorem 3.5.** *Under Conditions 3.2, 3.3, and 3.4, for any $T > 0$ such that $\|\widetilde{\beta}_t - \beta^*\|_{\widehat{\Gamma}} \geq \frac{\sqrt{L}}{\sqrt{s}}\|\widehat{\beta} - \beta^*\|_1$ for all $t \leq T$, we have*

$$\|\widetilde{\beta}_t - \beta^*\|_{\widehat{\Gamma}} \leq \gamma^{t-1}\|\widetilde{\beta}_1 - \beta^*\|_{\widehat{\Gamma}} + \frac{\epsilon_{\mathsf{st}}(N,D)}{1-\gamma},$$

*where $\gamma = \frac{c_\gamma \alpha_3}{\alpha_2}$ is the contraction factor ($c_\gamma > 0$ is an absolute constant) and*

$$\epsilon_{\mathsf{st}}(N,D) = \frac{2(1+12\alpha_3)\lambda_{\max}(\widehat{\Gamma})^{1/2}}{\alpha_2\sqrt{s}}\|\widehat{\beta} - \beta^*\|_1 + \frac{24\sqrt{s}}{\alpha_2\sqrt{\alpha_1}}\|\widehat{\Gamma}\beta^* - \widehat{\gamma}\|_\infty.$$

To interpret Theorem 3.5, recall

$$\|\widehat{\beta} - \beta^*\|_2 \lesssim_P \sqrt{s}\|\widehat{\Gamma}\beta^* - \widehat{\gamma}\|_\infty, \quad \|\widehat{\beta} - \beta^*\|_1 \lesssim_P s\|\widehat{\Gamma}\beta^* - \widehat{\gamma}\|_\infty,$$

where $\widehat{\beta}$ is the lasso estimator. Further, the prediction error of the lasso estimator is (up to a constant) $\frac{\sqrt{L}}{\sqrt{s}}\|\widehat{\beta} - \beta^*\|_1$, which (up to a constant) is exactly statistical precision $\epsilon_{\mathsf{st}}(N,D)$. Theorem 3.5 states that the prediction error of $\widetilde{\beta}_t$ decreases q-linearly to that of the lasso estimator. We emphasize that the convergence rate is linear despite the absence of strong convexity, which is usually the case when $N < D$. A direct consequence is that only logarithmically many iterations ensures a desired suboptimality, which stated in the following corollary.

**Corollary 3.6.** *Under the conditions of Theorem 3.5,*

$$T = \frac{\log\left(\epsilon - \frac{\epsilon_{\mathsf{st}}(N,D)}{1-\gamma}\right)^{-1} - \log\frac{1}{\epsilon_1}}{\log\frac{1}{\gamma}} \approx \log\frac{1}{\epsilon}$$

*iterations of the constrained quasi-Newton method, where $\epsilon_1 = \|\widehat{\beta}_1 - \beta^*\|_{\widehat{\Gamma}}$, is enough to produce an iterate whose prediction error is smaller than*

$$\epsilon > \max\left\{\frac{\lambda_{\max}(\widehat{\Gamma})^{1/2}}{\sqrt{s}}\|\widehat{\beta} - \beta^*\|_1, \frac{\epsilon_{\mathsf{st}}(N,D)}{1-\gamma}\right\} \approx \|\widehat{\beta} - \beta^*\|_{\widehat{\Gamma}}.$$

Theorem 3.5 is vacuous if the contraction factor $\gamma = \frac{c_\gamma \alpha_3}{\alpha_2}$ is not smaller than 1. To ensure $\gamma < 1$, it is enough to choose the sketch size $n$ so that $\frac{\alpha_3}{\alpha_2} < c_\gamma^{-1}$. Consider the "good event"

$$\mathcal{E}(\delta) := \left\{\alpha_2 \geq 1 - \delta, \alpha_3 \leq \frac{\delta}{2}\right\}. \tag{4}$$

If the rows of $S_t$ are sub-Gaussian, to ensure $\mathcal{E}(\delta)$ occurs with high probability, Pilanci and Wainwright show it is enough to choose

$$n > \frac{c_s}{\delta^2}\mathcal{W}\left(X_{\widehat{S}}(K(S) \cap \mathbb{S}^{d-1})\right)^2, \tag{5}$$

where $c_s > 0$ is an absolute constant and $\mathcal{W}(S)$ is the Gaussian-width of the set $S \subset \mathbb{R}^d$ [13].

**Lemma 3.7** (Pilanci and Wainwright (2014)). *For any sketching matrix whose rows are independent 1-sub-Gaussian vectors, as long as the sketch size $n$ satisfies* (5),

$$\mathbb{P}\big(\mathcal{E}(\delta)\big) \geq 1 - c_5 \exp\big(-c_6 n \delta^2\big),$$

*where $c_5, c_6$ are absolute constants.*

As a result, when the sketch size $n$ satisfies (5), Theorem 3.5 is non-trivial.

**Tradeoffs depending on sketch size.** We remark that the contraction coefficient in Theorem 3.5 depends on the sketch size. As the sketch size $n$ increases, the contraction coefficient decays and vice versa. Thus the sketch size allows practitioner to trade-off the total rounds of communication with the total amount of data (in bits) sent over the network. A larger sketch size results in fewer rounds of communication, but more bits per round of communication and vice versa. Recall [5] the communication cost of an algorithm is

$$\text{rounds} \times \text{overhead} + \text{bits} \times \text{bandwidth}^{-1}.$$

By tweaking the sketch size, users can trade-off rounds and bits, thereby minimizing the communcation cost of our approach on various distributed computing platforms. For example, the user of a cluster comprising commodity machines is more concerned with overhead than the user of a purpose-built high performance cluster [7]. In the following, we study the two extremes of the trade-off.

At one extreme, users are solely concerned by the total amount of data sent over the network. On such platforms, users should use smaller sketches to reduce the total amount of data sent over the network at the expense of performing a few extra iterations (rounds of communication).

**Corollary 3.8.** *Under the conditions of Theorem 3 and Lemma 3.7, selecting $d := \lfloor \tau N \rfloor$ features by SIS, where $\tau = cN^{-\theta}$ for some $c > 0$ and $\theta < 1 - 2\kappa$ and letting*

$$n > \frac{c_s(c_\gamma + 2)^2}{4} \mathcal{W}\big(\boldsymbol{X}_{\widehat{S}}(K(S) \cap \mathbb{S}^{d-1})\big)^2, \quad T = \frac{\log \frac{1}{\epsilon_{\mathsf{st}}(N,D)} - \log \frac{1}{\epsilon_1}}{\log 2}$$

*in Algorithm 1 ensures $\|\widetilde{\boldsymbol{\beta}}_T - \beta^*\|_{\widehat{\boldsymbol{\Gamma}}} \leq 3\epsilon_{\mathsf{st}}(N,D)$ with probability at least*

$$1 - c_4 T \exp\big(-c_2 n \delta^2\big) - C_2 \exp\big(-\frac{CN^{1-2\kappa}}{\log N}\big),$$

*where $c, c_\gamma, c_s, c_2, c_4, C, C_2$ are absolute constants.*

We state the corrollary in terms of the statistical precision $\epsilon_{\mathsf{st}}(N,D)$ and the Gaussian width to keep the expressions concise. It is known that the Gausssian width of the transformed tangent cone that appears in Corollary 3.8 is $O(s \log d)^{1/2}$ [13]. Thus it is possible to keep the sketch size $n$ on the order of $s \log d$. Recalling $d = \lfloor \tau N \rfloor$, where $\tau$ is specified in the statement of Theorem 3.1, and $\epsilon_{\mathsf{st}}(N,D) \leq \big(\frac{s \log D}{N}\big)^{\frac{1}{2}}$, we deduce the communication cost of the approach is

$$O(dnmT) = O\big(N(s \log d)m \log\big(\tfrac{N}{s \log D}\big)\big) = \widetilde{O}(mns),$$

where $\widetilde{O}$ ignores polylogarithmic terms. The takeaway is it is possible to obtain an $O(\epsilon_{\mathsf{st}}(N,D))$ accurate solution by sending $\widetilde{O}(mNs)$ bits over the network. Compared to the $O(mN^2)$ communication cost of DECO, we see that our approach exploits sparsity to reduce communication cost.

At the other extreme, there is a line of work in statistics that studies estimators whose evaluation only requires a single round of communication. DECO is such a method. In our approach, it is possible to obtain an $\epsilon_{\mathsf{st}}(N,D)$ accurate solution in a single iteration by choosing the sketch size large enough to ensure the contraction factor $\gamma$ is on the order of $\epsilon_{\mathsf{st}}(N,D)$.

**Corollary 3.9.** *Under the conditions of Theorem 3 and Lemma 3.7, selecting $d := \lfloor \tau N \rfloor$ features by SIS, where $\tau = cN^{-\theta}$ for some $c > 0$ and $\theta < 1 - 2\kappa$ and letting*

$$n > \frac{c_s(c_\gamma \epsilon_{\mathsf{st}}(N,D)^{-1} + 2)^2}{4} \mathcal{W}\big(\boldsymbol{X}_{\widehat{S}}(K(S) \cap \mathbb{S}^{d-1})\big)^2$$

*and $T = 1$ in Algorithm 1 ensures $\|\widetilde{\boldsymbol{\beta}}_T - \beta^*\|_{\widehat{\boldsymbol{\Gamma}}} \leq 3\epsilon_{\mathsf{st}}(N,D)$ with probability at least*

$$1 - c_4 T \exp\big(-c_2 n \delta^2\big) - C_2 \exp\big(-\frac{CN^{1-2\kappa}}{\log N}\big),$$

*where $c, c_\gamma, c_s, c_2, c_4, C, C_2$ are absolute constants.*

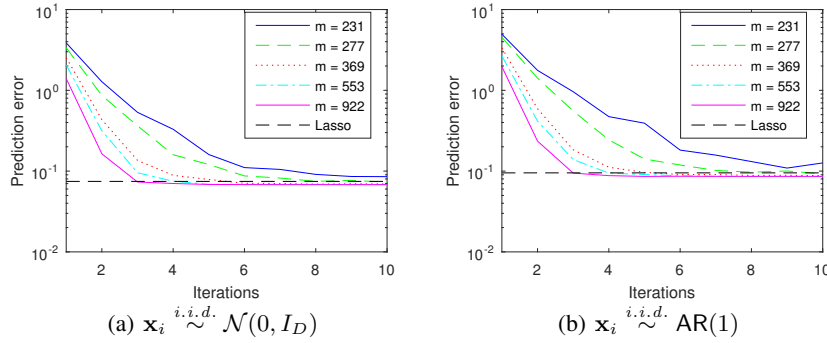

(a) $\mathbf{x}_i \overset{i.i.d.}{\sim} \mathcal{N}(0, I_D)$          (b) $\mathbf{x}_i \overset{i.i.d.}{\sim} \mathsf{AR}(1)$

Figure 1: Plots of the statistical error $\log \|\widetilde{\boldsymbol{X}}(\widehat{\boldsymbol{\beta}} - \beta^*)\|_2^2$ versus iteration. Each plots shows the convergence of 10 runs of Algorithm 1 on the same problem instance. We see that the statistical error decreases linearly up to the statistical precision of the problem.

Recalling
$$\epsilon_{\mathsf{st}}(N, D)^2 \approx \tfrac{s \log D}{N}, \quad \mathcal{W}\big(\boldsymbol{X}_{\widehat{S}}(K(S) \cap \mathbb{S}^{d-1})\big)^2 \approx s \log d,$$

we deduce the communication cost of the one-shot approach is
$$O(dnmT) = O\big(N^2 m \log\big(\tfrac{N}{s \log D}\big)\big) = \widetilde{O}(mN^2),$$

which matches the communication cost of DECO.

## 4 Simulation results

In this section, we provide empirical evaluations of our main algorithm SCREENANDCLEAN on synthetic datasets. In most of the experiments the performance of the methods is evaluated in terms of the prediction error which is defined as $\|\widetilde{\boldsymbol{X}}(\widehat{\boldsymbol{\beta}} - \beta^*)\|_2^2$. All the experiments are implemented in Matlab on a shared memory machine with 512 GB RAM with 4(6) core intel Xeon E7540 2 GHz processors. We use TFOCS as a solver for any optimization problem involved, e.g., step 5 in Algorithm 1. For brevity, we refer to our approach as SC in the rest of the section.

### 4.1 Impact of number of iterations and sketch size

First, we confirm the prediction of Theorem 3.5 by simulation. Figure 1 shows the prediction error of the iterates of Algorithm 1 with different sketch sizes $m$. We generate a random instance of a sparse regression problem with size 1000 by 10000 and sparsity $s = 10$, and apply Algorithm 1 to estimate the regression coefficients. Since Algorithm 1 is a randomized algorithm, for a given (fixed) dataset, its error is reported as the median of the results from 11 independent trials. The two subfigures show the results for two random designs: standard Gaussian (left) and AR(1) (right). Within each subfigure, each curve corresponds to a sketch size, and the dashed black line show the prediction error of the lasso estimator. On the logarithmic scale, a linearly convergent sequence of points appear on a straight line. As predicted by Theorem 3.5, the iterates of Algorithm 1 converge linearly up to the statistical precision, which is (roughly) the prediction error of the lasso estimator, and then it plateaus. As expected, the higher the sketch size is, the fewer number of iteration is needed. These results are consistent with our theoretical findings.

### 4.2 Impact of sample size $N$

Next, we evaluate the statistical performance of our SC algorithm when $N$ grows. For completeness, we also evaluate several competing methods, namely, lasso, SIS [6] and DECO [17]. The synthetic datasets used in our experiments are based on model (1). In it, $\boldsymbol{X} \sim \mathcal{N}(0, I_D)$ or $\boldsymbol{X} \sim \mathcal{N}(0, \Sigma)$ with all predictors equally correlated with correlation 0.7, $\epsilon \sim \mathcal{N}(0, 1)$. Similar to the setting appeared in [17], the support of $\beta^*$, $S$ satisfies that $|S| = 5$ and its coordinates are randomly chosen from $\{1, \ldots, D\}$, and
$$\beta_i^* = \begin{cases} (-1)^{\mathsf{Ber}(0.5)}\big(|(0,1)| + 5\big(\tfrac{\log D}{N}\big)^{1/2}\big) & i \in S \\ 0 & i \notin S. \end{cases}$$

We generate datasets with fixed $D = 3000$ and $N$ ranging from 50 to 600. For each $N$, 20 synthetic datasets are generated and the plots are made by averaging the results.

In order to compare with methods such as DECO which is concerned with the Lagrangian formulation of lasso, we modify our algorithm accordingly. That is, in step 5 of Algorithm 1, we solve

$$\widetilde{\boldsymbol{\beta}}_t \leftarrow \arg\min_{\beta \in \mathbb{R}^d} \frac{1}{2}\beta^T \widetilde{\boldsymbol{\Gamma}}_t \beta - (\widehat{\boldsymbol{\gamma}} - \widehat{\boldsymbol{\Gamma}}\widetilde{\boldsymbol{\beta}}_{t-1})^T \beta + \lambda \|\beta\|_1.$$

Herein, in our experiments, the regularization parameter is set to be $\lambda = 2\|\boldsymbol{X}^T\boldsymbol{\epsilon}\|_\infty$. Also, for SIS and SC, the screening size is set to be $2N$. For SC, we run it with sketch size $n = 2s\log(N)$ where $s = 5$ and 3 iterations. For DECO, the dataset is partitioned into $m = 3$ subsets and it is implemented without the refinement step. The results on two kinds of design matrix are presented in Figure 2.

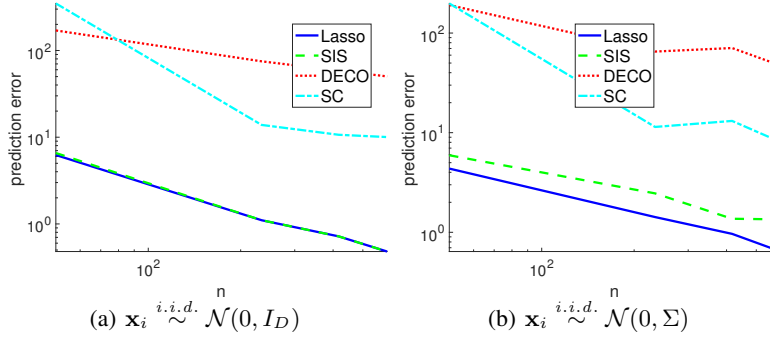

(a) $\mathbf{x}_i \overset{i.i.d.}{\sim} \mathcal{N}(0, I_D)$      (b) $\mathbf{x}_i \overset{i.i.d.}{\sim} \mathcal{N}(0, \Sigma)$

Figure 2: Plots of the statistical error $\log \|\widetilde{\boldsymbol{X}}(\widehat{\boldsymbol{\beta}} - \beta^*)\|_2^2$ versus $\log N$. In the above, (a) is generated on datasets with independent predictors and (b) is generated on datasets with correlated predictors. Besides our main algorithm SC, several competing methods, namely, lasso, SIS and DECO are evaluated. Here $D = 3000$. For each $N$, 20 independent simulated datasets are generated and the averaged results are plotted.

As can be seen, SIS achieves similar errors as lasso. Indeed, after careful inspection, we find out that when in the cases where predictors are highly correlated, i.e., Figure 2(b), usually less than 2 non-zero coefficients can be recovered by sure independent screening. Nevertheless, this doesn't deteriorate the accuracy too much. Moreover, SC's performance is comparable to both SIS and lasso as the prediction error goes down in the same rate, and SC outperforms DECO in our experiments.

Finally, in order to demonstrate that our approach is amenable to distributed computing environments, we implement it using Spark[1] on a modern cluster with 20 nodes, each of which has 12 executor cores. We run our algorithm on an independent Gaussian problem instance with size 6000 and 200,000, and sparsity $s = 20$. The screening size is 2400, sketch size is 700, number of iterations is 3. To show the scalability, we report the running time using $1, 2, 4, 8, 16$ machines, respectively. As most of the steps in our approach are embarrassingly parallel, the running time becomes almost half as we double the number of machines.

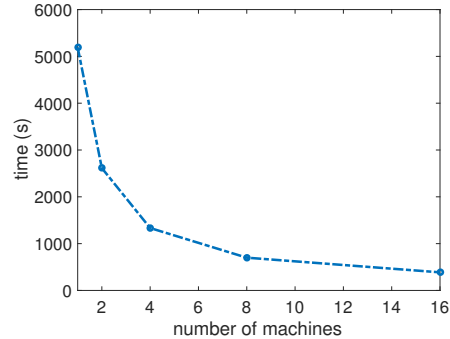

Figure 3: Running time of a Spark implementation of SC versus number of machines.

## 5 Conclusion and discussion

We presented an approach to feature-distributed sparse regression that exploits the sparsity of the regression coefficients to reduce communication cost. Our approach relies on sketching to compress the information that has to be sent over the network. Empirical results verify our theoretical findings.

**Acknowledgments.** We would like to thank the Army Research Office and the Defense Advanced Research Projects Agency for providing partial support for this work.

## Footnotes

[1] http://spark.apache.org/

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
