[Reviews · NeurIPS 2016]

Reviewer 1

Summary

The paper introduces an approach for feature distributed sparse linear regression, combining sure screening, sketching and a series of quadratic programs. Theoretical guarantees are provided on the iterates showing convergence to their error to that achieved by the (centralized) lasso solution. The impact of the sketch size on the number of rounds of communication and the amount of data sent over the network is discussed. Simulation experiments are provided to illustrate the theoretical results and evaluate the statistical error as a function of the sample size.

Qualitative Assessment

The paper presents some interesting contributions. The approach and theoretical results are promising. However the empirical evaluation could be more thorough to demonstrate the claims made, e.g - varying sketch size (and screening size) could be considered to quantify the tradeoff in communication rounds and data shared. - Various correlation settings for the features could be considered, with varying condition number for the selected features, to show resilience of the proposed approach. It would also be helpful to see results for varying sparsity and noise level. Timing results should be more systematically reported and compared with other approaches. Minor comments: Inconsistent notation: sample size denoted either as N and n. L53: missing reference

Confidence in this Review

2-Confident (read it all; understood it all reasonably well)


Reviewer 2

Summary

the authors provide a novel algorithm for the sparse regression problem when the features are distributed in multiple machines. The problem and the setting are interesting. As far as i can tell there are no theoretical bounds for the proposed algorithm. Additionally the experimental results are not very convincing.

Qualitative Assessment

-

Confidence in this Review

2-Confident (read it all; understood it all reasonably well)


Reviewer 3

Summary

The authors present an approach to distributed sparse regression based on sketching. The proposed method partitions the, high-dimensional, data rather than samples. The proposed method uses a two-step process called screening and cleaning. In the screening step, the overall number of features is reduced to d=O(N), where N is the number of samples, by removing irrelevant features. By using a screening method that has the sure screening property the authors ensure that relevant features are not removed. In the cleaning step the reduced regression problem is solved iteratively via a series of regularized quadratic programs (line 96). At each iteration, the current estimate \hat{y} is computed in a distributed manner, each node then sends to the central machine the correlations of the sketched features with current estimate \hat{y}. The central machine then updates the estimate of the regression coefficients \beta. This iterative process terminates after a predetermined number of iterations. The authors provide a theoretical analysis of the distributed algorithm showing that the prediction error of \beta convergences q-linearly to that of the lasso estimator. The authors further show that the number of iterations of the cleaning stage required to produce an iterate with a prediction error smaller than \epsilon depends on the sketch size n. As the communication costs of the proposed method also depend on n, the authors show that there is a trade-off between "the total rounds of communication with the total amount of data (in bits) sent over the network".

Qualitative Assessment

I find the work very interesting and of high quality. I would agree with the motivation put forward by the authors that data split by features rather than samples is an interesting setting to explore in cases where the data is high-dimensional. What is not entirely clear is the exact contribution of the paper hence my low score on question 6. Reading Pilanci and Wainwright's JMLR paper it would seem that the cleaning stage of this work is very close to being an application of the algorithm proposed there (Iterative Hessian Sketching) to a distributed setting. The authors point out some points of difference, lines 52-55, but I am not convinced that these really constitute a novel contribution. I hope the authors can elaborate further on this point. What are the practical differences of applying the proposed method as opposed to IHS, are there tangible advantages to using the proposed method? Obviously IHS would incur a much greater communication cost for the same number of iterations. However how is convergence affected in this case?

Confidence in this Review

2-Confident (read it all; understood it all reasonably well)


Reviewer 4

Summary

The paper addresses the problem of partitioning data by features instead of samples. The paper proposes using a screening process to reduce dimensions and then cleaning to fit the sparse linear model to the selected features. The contribution of this paper is to use sub-guassian sketches. By tweaking the sketch size, users can trade-off communication rounds/bits, making the algorithm applicable to various networking interconnects.

Qualitative Assessment

The paper addresses an important problem of feature partitioning using second order methods. They cleverly use sparsity of regression coefficients to reduce communication costs. I was wondering what are the CPU costs when trading off the number of bits in communication since sub-Gaussian sketches are computationally intensive and even for small 'd', the costs may be significant enough even though the network costs may be small. The paper tested different algorithms on only synthetic data sets and none of them are of very high dimensionality. It will be worthwhile seeing these results for example for D=10000.

Confidence in this Review

2-Confident (read it all; understood it all reasonably well)


Reviewer 5

Summary

This article proposes a "divide and conquer" framework for processing data with large dimension. In particular, it partitions the feature space rather than the sample space. The tool it relies on are the variable screening method and the sketch method. The algorithm first partitions the data column-wisely into multiple subsets and screening out the irrelevant variables. It then enters another cleaning stage on the remaining features with sketching. The authors compare their approach to other methods via examples.

Qualitative Assessment

The proposed method addressed an interesting problem for big data inference. However, the methodology is mostly combinations of existing algorithms, such as sure independence screening and sketching. In addition, the inferencing framework is almost identical to the one proposed in "A split-and-merge Bayesian variable selection approach for ultrahigh dimensional regression" (Song and Liang, 2014). Song and Liang (2014) also partitions the feature set and apply variable screening in the first step. The only difference is that they used a Bayesian variable selection in the second stage while the authors in this article uses sketching and iteratively fit the sparse regression with several communication step required. While the method might work well for some examples, I felt the contribution and novelty of this work is limited. Also, the experiment setting seems unfair. The authors compared their method to DECO without the refinement step, while the "clean" stage in the proposed algorithm serves just as a refinement step. I think a comparison of DECO with refinement will be more convincing.

Confidence in this Review

2-Confident (read it all; understood it all reasonably well)


Reviewer 6

Summary

This paper tackles the problem of sparse linear regression with distributed features. In the method, they combine the sure independence screening and sketching to achieve the goal.

Qualitative Assessment

The paper is poorly written. A few of the references are missing. The definition of the sub-Gaussian random variable is wrong.

Confidence in this Review

1-Less confident (might not have understood significant parts)